# Molecularly Targeted Therapies for Gastric Cancer. State of the Art

**DOI:** 10.3390/cancers13164094

**Published:** 2021-08-14

**Authors:** Rossella Reddavid, Simona Dagatti, Caterina Franco, Lucia Puca, Mariano Tomatis, Simona Corso, Silvia Giordano, Maurizio Degiuli

**Affiliations:** 1Department of Oncology, Università degli Studi di Torino, 10126 Torino, Italy; rossella.reddavid@unito.it (R.R.); simona.dagatti@unito.it (S.D.); caterinafranco@hotmail.it (C.F.); lucia.pucamed@gmail.com (L.P.); mariano.tomatis@gmail.com (M.T.); 2Surgical Oncology and Digestive Surgery Unit, San Luigi University Hospital, Regione Gonzole 10, Orbassano, 10043 Turin, Italy; 3Department of Oncology, University of Torino, 10060 Candiolo, Italy; simona.corso@ircc.it (S.C.); silvia.giordano@ircc.it (S.G.); 4Candiolo Cancer Institute, FPO-IRCCS, Strada Provinciale 142, Candiolo, 10060 Turin, Italy

**Keywords:** gastric cancer, molecular target therapy, chemotherapy, EGFR inhibitors, angiogenesis inhibitors, MET inhibitors

## Abstract

**Simple Summary:**

Despite recent advances in surgical techniques and in anticancer drugs, and the adoption of perioperative treatments mostly based on conventional chemotherapy, the prognosis of advanced and metastatic gastric cancer remains poor. In the last decade, the addition of molecular therapy did not show any significant survival advantage, and the first reports available documented an increase of the rate of severe adverse effects and related mortality. We conducted a literature search for randomized trials investigating novel molecular agents as compared to conventional chemotherapy. The outcomes were patients’ survival and the rates of tumor response and of severe adverse effects (SAE). Although we did not find an increase of SAE, the survival benefits of novel molecular therapies available to date for advanced and metastatic gastric cancer were rather unclear, mostly due to inaccurate patient selection, particularly concerning oncogene amplification and copy number.

**Abstract:**

Many phase III trials failed to demonstrate a survival benefit from the addition of molecular therapy to conventional chemotherapy for advanced and metastatic gastric cancer, and only three agents were approved by the FDA. We examined the efficacy and safety of novel drugs recently investigated. PubMed, Embase and Cochrane Library were searched for phase III randomized controlled trials published from January 2016 to December 2020. Patients in the experimental arm received molecular therapy with or without conventional chemotherapy, while those in the control arm had conventional chemotherapy alone. The primary outcomes were overall and progression-free survival. The secondary outcomes were the rate of tumor response, severe adverse effects, and quality of life. Eight studies with a total of 4223 enrolled patients were included. The overall and progression-free survival of molecular and conventional therapy were comparable. Most of these trials did not find a significant difference in tumor response rate and in the number of severe adverse effects and related deaths between the experimental and control arms. The survival benefits of molecular therapies available to date for advanced and metastatic gastric cancer are rather unclear, mostly due to inaccurate patient selection, particularly concerning oncogene amplification and copy number.

## 1. Introduction

Gastric cancer is one of the most frequent malignancies. It represents the fifth most frequent cancer worldwide (5.6%) and the fourth leading cause of cancer-related death (7.7%) with 768,793 deaths per year in 2020 [1].

Surgical resection with optimal lymphadenectomy is the only curative treatment in cases of AGC [2,3,4,5,6]. In recent decades, several perioperative and postoperative regimens of conventional CT have been investigated, and neoadjuvant treatment has been recommended as mandatory in several national guidelines, but the prognosis of stage III and IV GC remains poor [7,8,9,10]. In 2014, Cancer Genome Atlas Research Network paved the way for a new molecular classification of GC and documented the existence of four subtypes: EBV (9%), MSI (22%), CIN (50%), and GS (20%) [11]. The identification of these subtypes and the related signaling pathways provided a roadmap for GC patient stratification and promising strategies for targeted therapies. Trastuzumab was the first MT approved by the FDA and European Union for AGC; it was subsequently introduced as the standard of care for patients with locally or fAGC displaying HER2 overexpression/amplification [12]. In 2014, the FDA also approved the use of ramucirumab as monotherapy or in combination with paclitaxel for advanced and metastatic GC [13]. To date, only these two MTs (in addition to the antibody–drug conjugate trastuzumab deruxtecan) have been approved, although many other molecular targets have been identified in recent years. Indeed, the majority of phase III trials investigating novel molecular agents failed to demonstrate their efficacy, mostly due to inaccurate patient selection (particularly concerning driver gene amplification and copy number) and the lack of preclinical models supporting proof of concepts followed by structured trials. PDXs are helpful in validating and predicting the response to novel MTs, even though these models are unable to reproduce the same conditions and environmental characteristics of the donor tumor and very rarely allow metastatic dissemination [14]. For this purpose, PDOXs were recently introduced in GC preclinical research to better recapitulate the original cancer background [15].

In 2016, the Cochrane Collaborative Group published a systematic review with the aim of assessing the efficacy and safety of MTs available for the treatment of advanced and metastatic gastric cancer [16]. The authors identified 11 RCTs enrolling a total of 4014 patients with AGC who underwent conventional CT and MT or conventional CT alone. They concluded that the benefit of MTs on survival was unclear and pointed out a significant increase in side effects.

The present systematic review and meta-analysis aims to examine the efficacy and safety of novel MTs investigated in the years after publication of the Cochrane review.

## 2. Molecular Targets and Target Agents

### 2.1. Epidermal Growth Factor Receptor

EGFRs include four types of TKRs (HER1/EGFR, HER2, HER3, HER4) located on the cell surface. They play an important role, conveying messages to manage cell growth and differentiation.

#### 2.1.1. Anti-HER1

Many authors have demonstrated that approximately 30% of GCs show HER1 overexpression [17,18]. Two main monoclonal antibodies (cetuximab and panitumumab) that reduce HER1 activity by binding its extracellular domain have been identified. Moreover, cetuximab can stimulate the activity of the immune system against tumor cells [19]. Unfortunately, the heterogeneity of GC seems to affect the efficacy of cetuximab in most of these patients [20].

Gefitinib and erlotinib, two tyrosine kinase inhibitors, can also inactivate HER1 by binding its intracellular domain and blocking its kinase activity [21]. Unfortunately, phase II trials have shown that these therapies have limited efficacy [22,23]. Recently, Maron et al. and Corso et al. identified a subpopulation of GC patients presenting a high level of EGFR amplification, which is responsive to anti-EGFR drugs [24,25]. They also identified mechanisms of resistance to EGFR-targeted drugs, such as TKR activation, KRAS mutation/amplification, and TSC2 inactivation [25].

#### 2.1.2. Anti HER2

Several authors have shown a direct relationship between HER2 amplification (and the consequent overexpression of its receptor) and many types of tumors [26]. The HER2 gene is a proto-oncogene located on chromosome 17q21. The first drug binding HER2 was trastuzumab. In 2010, the ToGa trial documented the superiority of trastuzumab in combination with conventional chemotherapy compared with chemotherapy alone in terms of OS and DFS for patients with AGC [12]. Nevertheless, only a few patients with GC (less than 20%) gain a real advantage from trastuzumab.

In the past decade, several other anti-HER2 agents have been tested for GC treatment. Lapatinib is a dual kinase inhibitor that acts on EGFR (ErbB1) and HER2 (ErbB2) with the consequent downregulation of HER2 signaling [27].

Pertuzumab is an anti-HER2 monoclonal antibody that prevents heterodimerization between HER2 and other HER family members [28].

The efficacy of the combination of trastuzumab and pertuzumab has been investigated in the JACOB trial [29]. Despite the suggestion of treatment activity (a trend towards therapeutic activity for increasing PFS and the proportion of patients who achieved an objective response), adding pertuzumab to trastuzumab and chemotherapy did not significantly improve OS in patients with HER2-positive GC vs. placebo. However, a recent preclinical trial demonstrated that a subgroup of patients with hyperamplified (>8 gene copies) HER2 could strongly benefit from dual HER2 blockade therapy [30].

T-DM1 is an antibody–drug conjugate generated by the conjugation of trastuzumab and DM1, a tubulin inhibitor [31]. The action of this drug is characterized by two phases: first, the ADC ligates the extracellular domain of HER2; it is subsequently transferred intracellularly, releasing DM1 that proceeds to block microtubule polymerization. The GATSBY trial, a randomized, open-label, adaptive, phase II/III study investigating the efficacy of T-DM1 compared to taxane in patients with previously treated, HER2-positive AGC, has just been completed and will be analyzed in this review [32].

Trastuzumab deruxtecan (DS-8201) is an antibody–drug conjugate consisting of trastuzumab, a cleavable linker, and a cytotoxic topoisomerase I inhibitor. An open-label, randomized, phase II trial performed on HER2+ GC patients evaluated trastuzumab deruxtecan vs. chemotherapy and showed that treatment with trastuzumab deruxtecan led to significant improvements in response and OS compared with standard therapies [33].

### 2.2. Vascular Endothelial Growth Factor

VEGFs are proteins promoting blood vessel formation. Four types of VEGF (VEGF-A, VEGF-B, VEGF-C, and VEGF-D) have been identified, with three types of corresponding receptors (VEGFR-1, VEGFR-2, and VEGFR-3). Several studies have reported the fundamental role of these signaling proteins in new blood vessel formation and cancer cell proliferation [34]. Furthermore, VEGF expression has been found in approximately 40% of GC [35]. Bevacizumab is an anti-VEGF-A monoclonal antibody that inhibits circulating VEGF-A activity [36]. The efficacy of this monoclonal antibody has been widely documented in several solid tumor treatments [37,38,39] but bevacizumab is still under investigation for its benefit in GC. Some phase II/III trials proved its efficacy in association with conventional chemotherapy in AGC, while others did not report any clear benefits [40,41]. Furthermore, Shah et al. reported improved oncologic outcomes only in Caucasian patients compared to Asian patients, suggesting that the VEGF-A pathway in GC could be different among races [42].

Many trials have investigated the efficacy of VEGF TKR inhibitors (sunitinib and sorafenib), but no phase III trial has shown any survival benefits [43,44]. Finally, a monoclonal antibody blocking VEGFR-2 was successfully introduced for advanced solid malignancy treatment in 2010 (ramucirumab) [45]. A significant improvement in survival outcomes in patients with AGC submitted to second-line therapy with ramucirumab alone or in combination with paclitaxel was documented in two main phase III trials [46,47]. Interestingly, these two trials also highlighted significant differences in the VEGF-A pathway between Asian and non-Asian patients.

### 2.3. Mammalian Target of Rapamycin

mTOR is a serine/threonine protein kinase identified in mammalian cells with a leading role in controlling mechanisms of cell growth and proliferation. Human cancers can be characterized by hyperactivity or inactivity of the mTOR pathway, which plays a crucial role in maintaining tumor-modified phenotypes [48]. In 2008, Cejka et al. [49] demonstrated in vitro the efficacy of everolimus (RAD001) in inhibiting mTOR complex 1 (mTORC1, mTOR combined with the adaptor protein raptor) with consequent blockage of HIF-1α and VEGF. The authors concluded that everolimus, through the inhibition of mTORC1 in GC cells, could affect cancer proliferation and generate central tumor necrosis. Moreover, everolimus antitumor action is amplified by its association with metronomic cyclophosphamide.

### 2.4. Hepatocyte Growth Factor Receptor

HGFR, also known as c-MET, is a proto-oncogenic receptor tyrosine kinase that, after binding to hepatocyte growth factor, induces cell migration and proliferation, promotes mitosis, and inhibits apoptosis. C-MET overexpression and gene amplification are related to a poor prognosis [50,51].

Crizotinib (PF-02341066) is a tyrosine kinase inhibitor of the c-MET receptor and of the TKR anaplastic lymphoma kinase; it has been approved by the FDA for treatment of ALK-positive NSCLC patients. Okamoto et al. in 2012 stated that crizotinib “has pronounced effects on signal transduction and survival in gastric cancer cells with MET amplification” [52]. Phase II/III trials to evaluate crizotinib efficacy and safety in GC are ongoing.

Another promising agent targeting the HGF-cMET complex is rilotumumab. This human monoclonal antibody impairs the c-MET signaling pathway by binding to and inactivating its ligand HGF [53]. Clinical trials of this drug in GC (including two phase III trials) were halted due to a significant increase in mortality in the experimental arm (rilotumumab in combination with conventional chemotherapy) in one of these trials, but new investigations have begun.

Finally, ornatuzumab is a humanized monoclonal antibody that binds to the extracellular receptor of c-MET, counteracting its activation by HGF ligand [54]. METGastric, a phase III trial of onartuzumab plus standard first-line chemotherapy for HER2, was recently conducted in MET+ advanced GC. Results of this study will be discussed in this review.

Table 1 summarizes the disappointing results of phase II and III trials that target HER2, EGFR, VEGF, VEGFR, MET, mTOR, and others.

In Figure 1, targeted therapies and oncogenic pathways in gastric cancer are detailed.

### 2.5. Preclinical Trials

Preclinical trials have proved to be valuable tools to derive molecular information to better target GC for innovative MTs and stratify patients for clinical trials. The use of organoids, PDXs, and PDOXs in GC research showed interesting patient related tumor characteristics and cancer escape mechanisms. Several authors reported a strong relationship between higher levels of HER2 amplification/copy number and increased benefit of Trastuzumab in AGC [30,62]. More recently, a preclinical trial on PDXs allowed a TSC2 mutation leading to increased resistance to EGFR inhibition to be identified. The pharmacological inhibition of TSC2 was positively tested with everolimus, which was able to overcome the resistance and to reestablish the sensitivity to EGFR inhibition [25].

## 3. Materials and Methods

### 3.1. Inclusion and Exclusion Criteria

The articles included in this systematic review and meta-analysis were phase III RCTs with available abstracts and full texts in English. In the experimental arm of the trial, patients received a molecular agent with or without conventional CT, while in the control arm, they received a placebo or conventional CT alone. Trials containing immunotherapy were not considered.

Reviews, meta-analyses, letters to the editor, editorials, case reports, retrospective studies, and conference abstracts were excluded.

Only RCTs recruiting adult patients (>18 years) with histologically proven gastric adenocarcinoma, with or without metastasis, were included in this study.

### 3.2. Outcomes

The primary outcomes of this meta-analysis were OS and PFS.

The secondary outcomes were overall response rate according to RECIST criteria, QoL, and side effects evaluated with specific scores [63,64].

### 3.3. Search Strategy

A computerized literature search of PubMed, Embase, and the Cochrane Library Central Register of Controlled Trials databases was conducted in December 2020 covering a period from 1/1/2016 to 9/12/2020, using combinations of free-text words and Medical Subject Headings (MeSH)/EMTREE terms: (“Stomach Neoplasms”[Mesh] OR ((stomach[tiab] OR gastric[tiab] OR esophago-gastr*[tiab] OR gastro-esophag*[tiab] OR gastroesophag*[tiab] OR oesophagogastr*[tiab] OR oesophago-gastr*[tiab] OR gastro-oesophag*[tiab]) AND (cancer*[tiab] OR tumor*[tiab] OR tumour*[tiab] OR neoplas*[tiab] OR carcinoma*[tiab] OR adenocarcinoma*[tiab] OR malignan*[tiab]))) AND (“Molecular Targeted Therapy”[Mesh] OR targeted-therap*[tiab] OR targeting-therap*[tiab] OR target-therap*[tiab] OR therapy-targeting[tiab] OR therapies-targeting[tiab] OR targeted-molecular[tiab] OR target-molecular[tiab] OR molecular-therap*[tiab] OR “Antibodies, Monoclonal”[Mesh] OR trastuzumab[tiab] OR “Lapatinib”[Mesh] OR lapatinib[tiab] OR cetuximab[tiab] OR panitumumab[tiab] OR “nimotuzumab”[Supplementary Concept] OR nimotuzumab[tiab] OR bevacizumab[tiab] OR “ramucirumab”[Supplementary Concept] OR ramucirumab[tiab] OR “apatinib”[Supplementary Concept] OR apatinib[tiab] OR “regorafenib”[Supplementary Concept] OR regorafenib[tiab] OR “rilotumumab”[Supplementary Concept] OR rilotumumab[tiab] OR “onartuzumab”[Supplementary Concept] OR onartuzumab[tiab] OR “Everolimus”[Mesh] OR everolimus[tiab] OR “zolbetuximab”[Supplementary Concept] OR claudiximab[tiab] OR zolbetuximab[tiab] OR “andecaliximab”[Supplementary Concept] OR andecaliximab[tiab] OR “Erlotinib Hydrochloride”[Mesh] OR erlotinib[tiab] OR “Gefitinib”[Mesh] OR gefitinib[tiab] OR”Sunitinib”[Mesh] OR sunitinib[tiab] OR “Sorafenib”[Mesh] OR sorafenib[tiab] OR “cediranib”[Supplementary Concept] OR cediranib[tiab] OR “GSK 1363089”[Supplementary Concept] OR foretinib[tiab] OR “Crizotinib”[Mesh] OR crizotinib[tiab] OR “marimastat”[Supplementary Concept] OR marimastat[tiab] OR prinostat[tiab] OR “AZD4547”[Supplementary Concept] OR AZD4547[tiab] OR AZD-4547[tiab] OR “brivanib”[Supplementary Concept] OR brivanib[tiab] OR “Vorinostat”[Mesh] OR vorinostat[tiab] OR “catumaxomab”[Supplementary Concept] OR catumaxomab[tiab] OR antibody-drug*[tiab] OR monoclonal-antibod*[tiab] OR “Protein Kinase Inhibitors”[Mesh] OR “Angiogenesis Inhibitors”[Mesh] OR “Matrix Metalloproteinase Inhibitors”[Mesh] OR “Histone Deacetylase Inhibitors”[Mesh] OR “ErbB Receptors”[Mesh] OR HER2[tiab] OR erbB-2[tiab] OR erbB2[tiab] OR erbB-1[tiab] OR erbB1[tiab] OR epidermal-growth-factor-receptor*[tiab] OR EGFR[tiab] OR EGF-receptor*[tiab] OR “Receptors, Vascular Endothelial Growth Factor”[Mesh] OR VEGF[tiab] OR vascular-endothelial-growth-factor-receptor*[tiab] OR VEGF-A[tiab] OR VEGFA[tiab] OR VEGFR[tiab] OR VEGFR-2[tiab] OR VEGFR2[tiab] OR VEGFR1[tiab] OR VEGFR-1[tiab] OR tyrosine-kinase[tiab] OR RTK[tiab] OR TIE2[tiab] OR TIE-2[tiab] OR “Proto-Oncogene Proteins c-met”[Mesh] OR c-MET[tiab] OR “Hepatocyte Growth Factor”[Mesh] OR hepatocyte-growth-factor[tiab] OR HGF[tiab] OR mammalian-target-of-rapamycin[tiab] OR mTOR[tiab] OR “CLDN18 protein, human”[Supplementary Concept] OR claudin-18*[tiab] OR anti-claudin[tiab] OR matrix-metalloproteinase*[tiab] OR MMPs[tiab] OR MMP-9[tiab] OR MMP9[tiab] OR histone-deacetylase[tiab]) AND ((“Randomized Controlled Trial”[Publication Type] OR “Controlled Clinical Trial”[Publication Type] OR random*[tiab] OR trial[tiab] OR placebo[tiab] OR groups[tiab] OR RCT[tiab] OR CCT[tiab] OR NCT0*[tiab] OR NCT1*[tiab] OR NCT2*[tiab] OR NCT3*[tiab] OR NCT4*[tiab] OR NCT5*[tiab] OR NCT6*[tiab] OR NCT7*[tiab] OR NCT8*[tiab] OR NCT9*[tiab] OR phase-1[tiab] OR phase-I[tiab] OR phase-2[tiab] OR phase-II[tiab] OR phase-3[tiab] OR phase-III[tiab] OR placebo[tiab]) NOT (“Animals”[Mesh] NOT “Humans”[Mesh])) AND (“2015/01/01”[Date-Entry]: “2020/12/09”[Date-Entry]).

The review was conducted according to the PRISMA guidelines for systematic reviews [65].

### 3.4. Data Selection

Three reviewers (S.D., C.F., and L.P.) independently screened the titles and abstracts and identified the appropriate studies based on the selection criteria.

In addition, a fourth author (R.R.) reviewed the selected abstracts. Subsequently, authors obtained the full texts to verify their appropriateness.

Disagreements between reviewers were resolved by repeated examination of the original articles and discussions within the team.

### 3.5. Quality Assessment

The quality of the included studies was evaluated by two independent reviewers (S.D. and C.F.) with the application of the Cochrane risk-of-bias tool for randomized trials (*RoB 2*) [66].

The selection of reported results, measurement of outcomes, missing outcome data, and deviation from the intended interventions and randomization processes were assessed for each trial.

### 3.6. Statistical Analysis

R software (version 4.0.5, R Foundation for statistical computing, Vienna, Austria) was used for pooling data and statistical analysis. For time-to-event outcomes (OS, PFS) and for severe adverse effects, we combined data using the generic inverse variance method presenting measurements of treatment effects as hazard ratios (HRs) and 95% confidence intervals (CIs). As in the 2016 Cochrane review, as the design of the agents of interest is based on a different mechanism (targeting different pathways), we used a random-effects model for primary analyses. Tests for heterogeneity were conducted using the Chi^2^ test. We adopted the *I²* statistic to estimate the total variation across studies due to heterogeneity [67]. If high levels of heterogeneity (*I*² > 50%) for primary outcomes were found, we explored possible sources using subgroup analyses. We did not perform tests for subgroup differences owing to the limited number of trials involved in each molecular prognostic biomarker subgroup.

## 4. Results

### 4.1. Literature Searches

The literature review and trial selection are detailed in Figure 2, based on PRISMA guidelines [65]. We conducted the search on the main electronic databases (950 articles found in MEDLINE, 4051 in EMBASE, and 1211 in CENTRAL) from 1 January 2015 to 9 December 2020 in collaboration with “Biblioteca Federata di Medicina, Università degli studi di Torino”. A total of 6212 papers were identified and subsequently deduplicated, resulting in 4634 included studies. After the first screening, 4497 studies were excluded because they did not meet inclusion criteria. An additional 114 articles were excluded because they were phase II trials or subgroup analysis-based studies. The remaining 23 articles were carefully analyzed, and 14 were removed. Reasons for exclusions are summarized in Figure 2. Subsequently, we excluded another article due to the inclusion of its data in the previously published Cochrane review [58]. Although one of the remaining eight trials was available only as an abstract, its detailed data and final findings were reported both in an American Society of Clinical Oncology presentation and on the *ClinicalTrials.gov* website; therefore, this study was not excluded [68].

Finally, 8 randomized controlled phase III trials with a total of 4223 enrolled patients were included in the present systematic review [32,68,69,70,71,72,73,74].

### 4.2. Risk of Bias in the Included Studies

The risk of bias in the included RCTs as calculated with the RoB2 tool is detailed in Figure 3.

The overall analysis resulted in half of the included trials showing a low risk of bias for all items [32,68,70,73], while some concerns were registered in one domain only in each of the remaining four studies [69,71,72,74].

#### 4.2.1. Study Characteristics

The main features of the enrolled trials are detailed in Table 2. Overall, more than half of the patients (76%) did not receive any previous line of chemotherapy, 14.7% of them were given only one line, and 6.4% and 3% were provided with two and three lines, respectively, before being included in the RCTs. Most of the trials evaluated OS as the primary endpoint, whereas three studies analyzed PFS.

In contrast to the other RCTs, Cunningham et al. [71] designed a study in a perioperative setting, also enrolling patients in early stages. However, generally, the patients included in this systematic review mostly had locally advanced, recurrent, or metastatic malignancies.

All selected trials analyzed both gastric and EGJ cancers; moreover, two of these trials also enrolled patients with esophageal malignancies [70,71].

The studies evaluated heterogeneous types of MTs with different targets: three of them used VEGFR targeting agents (apatinib [69], bevacizumab [71], ramucirumab [72]), two trials focused on c-MET inhibiting agents (onartuzumab [74] and rilotumumab [70]), one administered trastuzumab plus emtasine (anti-HER2) [32], and the remaining two studies investigated everolimus (anti-mTOR) [73] and andecaliximab (anti-MMP9) [68].

The majority of RCTs analyzed the efficacy of MT in combination with conventional CT compared to conventional treatment alone, with or without placebo, while the GATSBY [32] study compared MT alone versus conventional therapy. Curiously, the study by Li et al. compared the efficacy and safety of MT alone with those of placebo alone [69].

Two RCTs were terminated prematurely due to negative results [70,74]. Notably, the RILOMET-1 study was halted due to a significantly higher number of deaths in the experimental arm than in the control arm during a planned interim safety analysis.

#### 4.2.2. Survival Outcomes

All included RCTs analyzed both OS and PFS; results are detailed in Table 3. The median follow-up duration was available for seven of eight trials since the study by Li et al. [69] did not report follow-up information. It was 15.9 months (range, 6.2–39.1 months) for the experimental group and 15.2 months (range, 5.6–36.2 months) for the control group.

The meta-analysis showed that the global OS after targeted therapy was comparable to that after conventional therapy, with an HR of 0.99 (95%CI: 0.84; 1.16; *p* = 0.867, *I**^2^*** = 62%) (Figure 4).

Subsequently, OS was assessed considering 2 MT subgroups (Appendix A) according to the main categories of TKR inhibitors (VEGFR or c-MET inhibitors) administered to patients. This analysis confirmed the absence of a significant difference in survival between patients treated with a particular type of MT and those treated with conventional CT or placebo. In a total of 2942 patients, a meta-analysis of PFS was carried out using individual patient-level trial data. Similar to the OS findings, the use of MT did not show any improvement in PFS compared to conventional therapy or even to no treatment (HR 0.88, 95%CI: 0.68; 1.14, *p* = 0.286, *I**^2^* = 84%) (Figure 5).

Furthermore, the MT subgroup analysis (inhibitors of VEGFR vs. inhibitors of c-MET) confirmed the findings of the overall analysis (Appendix A).

#### 4.2.3. Secondary Outcomes

##### Overall Response Rate

Seven of the eight studies reported data about the ORR based on RECIST criteria (Table 4). The majority of these trials did not find a significant difference in ORR between the experimental and control groups. The RILOMET-1 study reported even a significantly better ORR in the control group [70], while the recent GAMMA-1 study registered a slightly higher ORR in the experimental arm (*p* = 0.049) [68].

##### Quality of Life

Only two RCTs evaluated patients’ QoL with the application of the EORTC QLQ-C30 global health status scale [69,72]. The QLQ-C30 response rate was high in every questionnaire domain in both studies, without any significant difference between the two groups.

##### Serious Adverse Effects

Finally, we proceeded to analyze the safety of the experimental arm compared to that of the control arm in terms of emergent SAE (grade ≥ 3) and SAE-related deaths. All of the articles described the occurrence of SAE. However, the meta-analysis of the available data showed that MT did not increase the number of SAEs compared with conventional treatment (HR 0.96, 95%CI: 0.78; 1.19, *I**^2^*** = 23%) (Figure 6).

The number of adverse events with fatal outcomes was detailed in seven of the eight included studies, as the trial by Li et al. [69] did not mention these data. As with the incidence of SAE, the administration of MT with or without conventional CT did not increase the rate of treatment-related deaths (HR 1.02, 95%CI: 0.82; 1.25, *I**^2^* = 0%) (Figure 6). Only the RCT by Catenacci et al. [70], investigating the safety and efficacy of rilotumumab (anti-cMET agent), was prematurely stopped due to a higher proportion of fatal adverse events, mostly due to disease progression, in the experimental arm than in the control arm. We used the fixed-effect model according to the absence of significant heterogeneity in both meta-analyses.

## 5. Discussion

GC is still characterized by a poor prognosis, particularly in cases of metastatic or recurrent disease and in locally advanced stages. The identification and introduction of effective and safe molecular therapies in clinical practice lag behind other malignancies, such as lung and breast cancers. To the best of our knowledge, this is the most recent systematic review and meta-analysis of emergent targeted therapies for GC.

Unfortunately, our findings showed that molecular therapies do not provide a clear survival benefit compared to conventional CT in the case of advanced or metastatic GC.

In 2016, the Cochrane group published the largest systematic review and meta-analysis investigating the survival benefit of MTs for GC patients, with or without conventional treatment. The Cochrane authors identified 11 RCTs (phase II and III studies), and the conclusion was “Adding molecular-targeted treatment to chemotherapy may have a small effect on survival and on stopping further development of the disease, compared with chemotherapy alone, but the evidence is of low quality”.

In the past five years, only eight new phase III RCTs have been conducted.

Most of these studies failed to demonstrate the superiority of MT with or without conventional CT compared with conventional treatment alone or with placebo in terms of survival outcomes. Moreover, two of these eight trials were terminated prematurely. The METGastric Phase III trial was stopped early because of negative results reported in a concomitant Phase II study that concluded: “The addition of onartuzumab to mFOLFOX6 in gastric cancer did not improve efficacy in an unselected population or in a MET immunohistochemistry-positive population” [74,75]. The RILOMET-1 was interrupted prematurely because a safety control committee found more deaths in the experimental arm than in the control arm during a planned interim analysis of safety and survival outcomes [70].

The RCT published by Li was the only positive study; it reported a clear survival benefit in patients with GC treated with apatinib (a VEGFR2 inhibitor) compared with those receiving a placebo in terms of both OS (7.6 vs. 5.0 months, *p* = 0.0027) and PFS (2.8 vs. 1.9 months, *p* < 0.001), with an acceptable SAE rate [69]. Accordingly, in 2014, the China Food and Drug Administration approved the use of apatinib as a third-line treatment for metastatic GC.

Despite this positive report, the overall meta-analysis did not show any significant differences in OS and PFS between the experimental (MT) and control arms.

Furthermore, the subgroup analysis according to the type of MT administered (VEGFR or c-MET inhibitors) failed to show a significant prolongation of OS and/or PFS in the experimental arm. Notably, our results may have been unable to identify significant differences between the two arms due to the high heterogeneity found among the included studies. On account of this statistical bias, we conducted two further meta-analyses matching our OS and PFS findings with those reported in the Cochrane review [76,77,78,79,80,81] (Figure 7a,b; Appendix A). Regrettably, these new cumulative analyses maintained high heterogeneity and could not document any survival advantage when MT was added to conventional treatment or administered alone compared to conventional CT or to a placebo.

Most of the included trials reported no differences in the ORR evaluated according to RECIST criteria [63] between the two treatment arms, with the exception of the RILOMET-1 study [70], which registered a significantly worse response in the experimental arm, and in the GAMMA-1 trial [68], which, on the contrary, reported a significantly better result in the MT group.

Quality of life was mentioned only in the study by Li et al. [69] and in the RAINFALL study [72] without any significant differences between the two groups.

Finally, the number of serious adverse effects and SAE-related deaths did not increase in the experimental arm. Additionally, the analysis of secondary outcomes confirmed that, to date, the supposed advantage of the administration of MT vs. conventional CT alone is unclear.

In addition, most of the investigated targeted therapies available to date are very expensive; therefore, it is mandatory to evaluate the cost-effectiveness as well. In 2017, Chen et al. [82] evaluated the relationship between the efficacy and the costs of apatinib as a third-line treatment in metastatic GC and concluded that this type of treatment is not cost-effective at all, while another author stated that apatinib is likely to be cost-effective only for patients with solid insurance [83]. Other authors analyzed the cost-effectiveness ratio of ramucirumab + paclitaxel as a second line treatment in AGC as proposed by Wilke et al. [47], concluding that this regimen was cost-ineffective and suggesting that its indirect charges to society be considered [84,85].

Finally, although three MTs have been approved by the FDA (trastuzumab, trastuzumab–deruxtecan, and ramucirumab) and a fourth one by the China Food and Drug Administration (apatinib), most phase III RCTs assessing novel molecular agents failed to demonstrate a survival advantage over conventional treatments. Consistent with the literature, we found four possible reasons for these negative results.

First, only in recent times has GC undergone wide investigational programs from a molecular perspective, which has highlighted the importance of patient selection because of the high number of molecular mutations found in GC [86]. Indeed, several molecular alterations characterizing GC subtypes have been identified and analyzed in the past decade, as in the case of CIN tumors, which manifest the most frequent TKR amplifications, and in the case of 80% of EBV tumors, which display *PIK3CA* mutations [87].

Second, GC is often characterized by a high grade of heterogeneity, both inside the primary tumor and in distant metastases. Several studies clearly demonstrated the intratumoral heterogeneous pattern of HER2 and c-MET expression [88,89]. Some authors have suggested inactivating alterations to the phylogenetic tree trunk because they promote cancer growth and are present in every tumor cell [90]. Unfortunately, no trunk mutations have been discovered in GC.

Third, several preclinical trials have recently documented a strict relationship between c-MET amplification and copy number and the response grade to anti-MET therapies [91,92] and that c-MET expression alterations are found in only 2% of GCs. However, in clinical trials investigating anti-MET agents, no patient selection was done. This could be one of the reasons for RILOMET-1 and METGastric trial failure.

Finally, many studies have shown different escape mechanisms of cancer cells that could shorten the duration of or even nullify the response to targeted therapies [93,94]. For example, c-MET-addicted GC could overcome c-MET blockade through HER family receptor expression activation. Recently, Apicella et al. showed that combined molecular therapy with anti-MET/EGFR leads to a complete and durable response [91]. For this reason, PDX and PDOX are valuable preclinical tools in validating new targeted therapies tailored to patients’ cancer molecular expression [14,15,95].

## 6. Conclusions

The results of this systematic review and meta-analysis showed that despite their newly documented safety, the molecular therapies available to date for advanced and metastatic gastric cancer do not present clear survival benefits. These unfavorable results are mostly related to inadequate patient selection. Targeted therapies are promising treatments for patients with locally advanced, metastatic, or recurrent gastric cancer as they are for other types of tumors. However, their clinical validation requires accurate patient selection, particularly related to driver oncogene amplification and copy number, and it should take into account preclinical models investigating cancer heterogeneity and escape mechanisms.

## Figures and Tables

**Figure 1 cancers-13-04094-f001:**
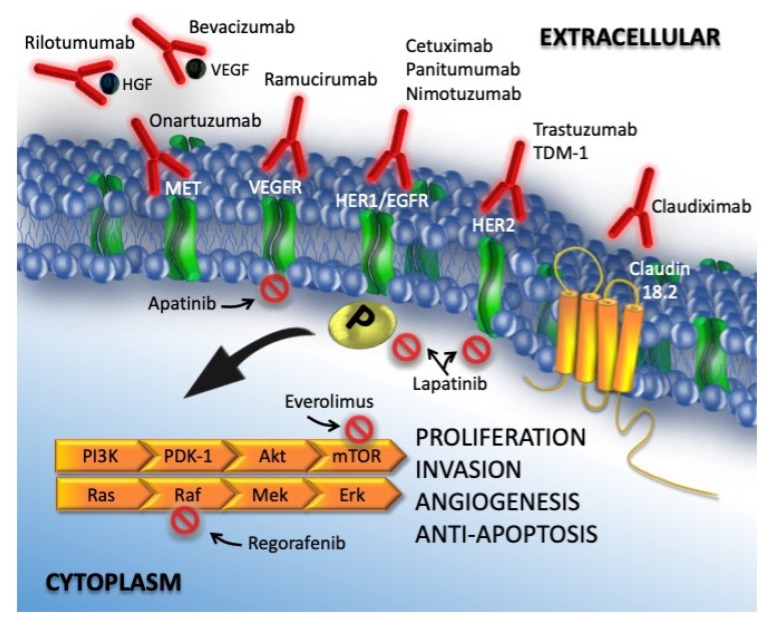
Targeted therapy and oncogenic pathways in gastric cancer. *Activation of ERK-AMP KINASE*: ligand binding to a growth factor receptor activates the small GTP-binding RAS protein, which interacts with RAF protein kinase. RAF phosphorylates and activates MEK (MAP kinase or ERK kinase), which then activates ERK (extracellular signal-regulated kinase) by phosphorylation of tyrosine and threonine residues. Activated ERK translocates into the nucleus where it phosphorylates the Elk-1 transcription. *PI3K/AKT/MTOR Pathway*: PI3K/AKT/MTOR signaling constitutes an important pathway that consists of two steps: phosphatidylinositol 3-kinase (PI3K) and its downstream molecule serine/threonine protein kinase B (PKB; also known as AKT). The PI3/AKT/mTOR pathway is stimulated by RTK and cytokine receptor activation. Tyrosine residues are then phosphorylated and provide anchor sites for PI3K translocation to the membrane, thus participating in the transduction of various extracellular matrix molecules and cytokines, including mTOR, a serine/threonine protein kinase and a member of the PI3K-associated kinase protein family.

**Figure 2 cancers-13-04094-f002:**
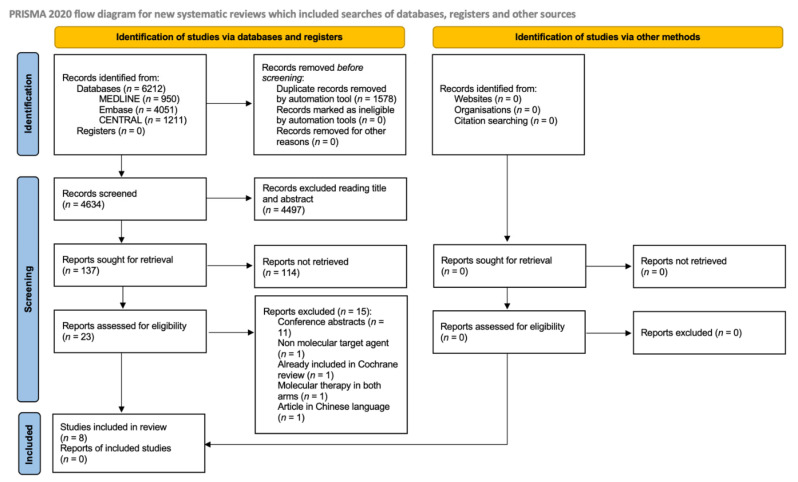
Prisma flow diagram.

**Figure 3 cancers-13-04094-f003:**
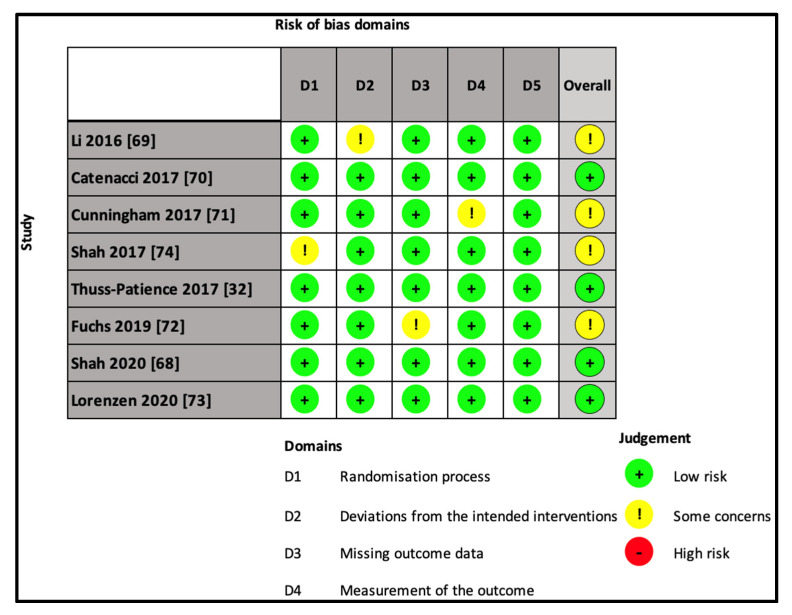
Risk of bias. To assess the risk of bias of each included study, the revised version of the Cochrane tool (RoB 2) was employed. The RoB 2 tool is structured into domains through which bias might be introduced into the result. These domains were identified based on both empirical evidence and theoretical considerations.

**Figure 4 cancers-13-04094-f004:**
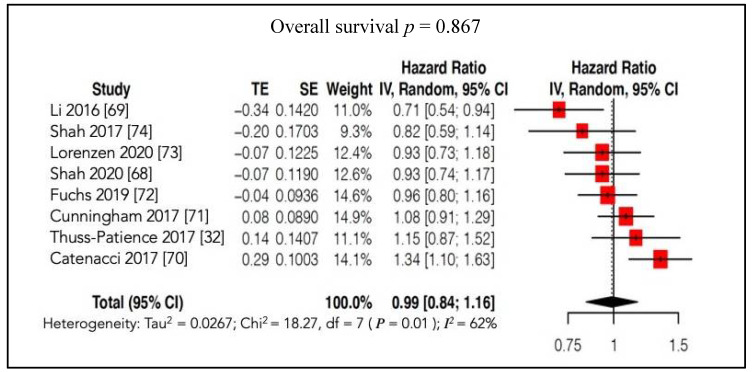
Forest plot of comparison: molecular-targeted therapy alone/plus chemotherapy versus chemotherapy alone/placebo. Main analyses; outcome: overall survival.

**Figure 5 cancers-13-04094-f005:**
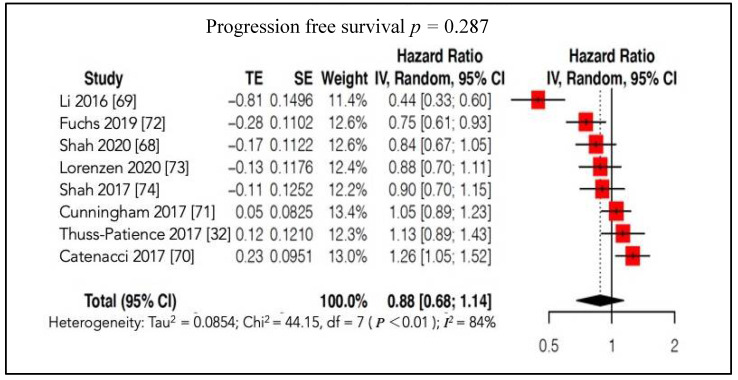
Forest plot of comparison: molecular-targeted therapy alone/plus chemotherapy versus chemotherapy alone/placebo. Main analyses; outcome: progression free survival.

**Figure 6 cancers-13-04094-f006:**
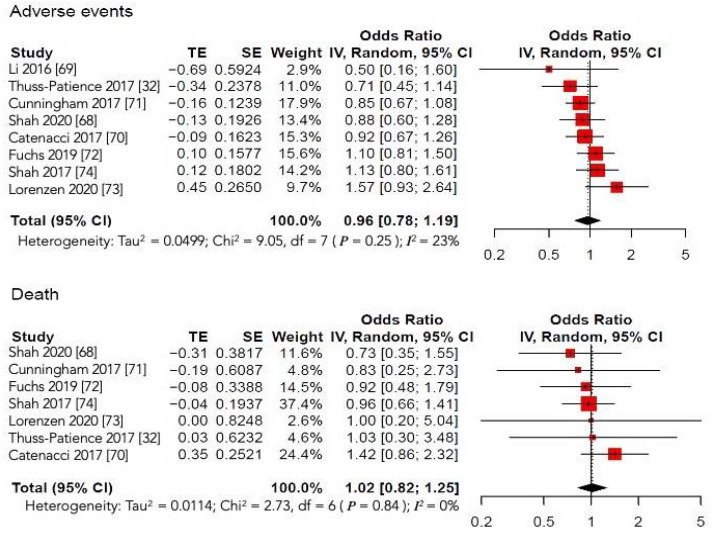
Forest plots of comparison: molecular-targeted therapy alone/plus chemotherapy versus chemotherapy alone/placebo. Secondary analyses; outcome: serious adverse effects and related-deaths.

**Figure 7 cancers-13-04094-f007:**
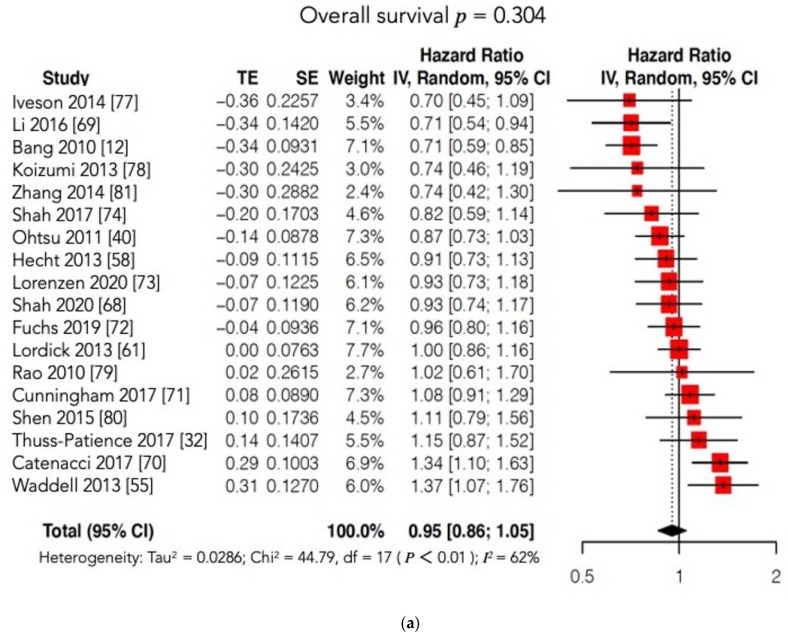
(**a**) Forest plot of comparison: molecular-targeted therapy alone/plus chemotherapy versus chemotherapy alone/placebo. Main analyses; outcome: overall survival (data from Cochrane and present review pooled). (**b**) Forest plot of comparison: molecular-targeted therapy alone/plus chemotherapy versus chemotherapy alone/placebo. Main analyses; outcome: progression free survival (data from Cochrane and present review pooled).

**Table 1 cancers-13-04094-t001:** Results of phase II and III trials. This table summarizes recent phase II and III RCTs investigating novel molecular agents’ survival outcomes. Unfortunately, most of these trials did not show any overall and progression free survival advantages as compared to conventional chemotherapy (red dot). Positive and partially positive studies have been pointed out with green and orange dot, respectively.

Trial, Year	EXP Arm	CTR Arm	Molecular Target	Nr Total Pts (EXP/CTR)	Treatment Line	Phase	Median OS (Months)	Median PFS (Months)	Results
REAL-3 [55], 2009	EOC + Panitumumab	EOC	EGFR	553	I	III	11.3 CTR arm8.8 EXP arm95%CI: 1.07–1.76 *p* = 0.013 HR = 1.37	7.4 CTR arm6.0 EXP arm95%CI: 0.98–1.52 *p* = 0.068 HR = 1.22	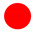
AVAGAST [40], 2012	XP + Bevacizumab	XP + Placebo	VEGF	774	I	III	10.1 CTR arm12.1 EXP arm95%CI: 0.73–1.03 *p* = 0.1002 HR = 0.87	5.3 CTR arm6.7 EXP arm95%CI: 0.68–0.93 *p* = 0.0037 HR = 0.80	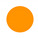
FAST [56], 2012	EOX + Claudiximab	EOX	Claudin 18.2	161	I	II	8.4 CTR arm13.4 EXP arm95%CI: 0.36–0.73 *p* < 0.001 HR = 0.51	4.8 CTR arm7.9 EXP arm95%CI: 0.31–0.70*p* = 0.0001 HR = 0.47	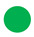
INTEGRATE [57], 2012	Regorafenib	Placebo	VEGF, RET, RAF	147	IIIII	II	4.5 CTR arm5.3 EXP arm95%CI: 0.51–1.08 *p* = 0.147 HR = 0.74	0.9 CTR arm2.6 EXP arm95%CI: 0.28–0.59*p* < 0.001 HR = 0.4	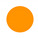
ENRICH (NCT01813253), 2013	Irinotecan + Nimotuzumab	Irinotecan	EGFR	400	II	III	NO RESULT POSTED	NO RESULT POSTED	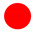
LOGiC [58], 2013	XELOX + Lapatinib	XELOX + Placebo	HER2	545	I	III	10.5 CTR arm12.2 EXP arm95%CI: 0.73–1.12 *p* = 0.3492 HR = 0.91	5.4 CTR arm6.0 EXP arm95%CI: 0.68–1 *p* = 0.0381 HR = 0.82	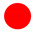
JapicCTI-090849 [59], 2014	Irinotecan + Nimotuzumab	Irinotecan	EGFR	83	II	II	7.7 CTR arm8.4 EXP arm95%CI: 0.618–1.599 *p* = 0.9778 HR = 0.994	2.9 CTR arm2.4 EXP arm95%CI: 0.516–1.435 *p* = 0.5668 HR = 0.860	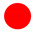
RAINBOW [47], 2014	Paclitaxel + Ramucirumab	Paclitaxel + Placebo	VEGFR2	665	II	III	7.36 CTR arm9.63 EXP arm95%CI: 0.678–0.962 *p* = 0.0169 HR = 0.807	2.86 CTR arm4.4 EXP arm95%CI: 0.536–0.752 *p* < 0.0001 HR = 0.635	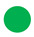
REGARD [46], 2014	Ramucirumab	Placebo	VEGFR2	355	II	III	3.8 CTR arm5.2 EXP arm95%CI: 0.603–0.998 *p* = 0.047 HR = 0.776	1.3 CTR arm2.1 EXP arm95%CI: 0.376–0.620 *p* < 0.0001 HR = 0.483	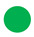
ToGA [12], 2014	FP/XP + Trastuzumab	FP/XP	HER2	594	I	III	11.1 CTR arm13.8 EXP arm95%CI: 0.60–0.91 *p* = 0.0046 HR = 0.74	5.5 CTR arm6.7 EXP arm95%CI: 0.59–0.85 *p* = 0.0002 HR = 0.71	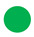
TyTAN [27], 2014	PTX + Lapatinib	PTX	HER2	261	II	III	8.9 CTR arm11 EXP arm95%CI: 0.64–1.11 *p* = 0.1044 HR = 0.84	4.4 CTR arm5.4 EXP arm95%CI: 0.63–1.13 *p* = 0.2241 HR = 0.85	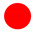
GRANITE-1 [60], 2015	Everolimus	Placebo	mTOR	656	IIIII	III	4.34 CTR arm5.39 EXP arm95%CI: 0.75–1.08 *p* = 0.1244 HR = 0.90	1.41 CTR arm1.68 EXP arm95%CI: 0.56–0.78 *p* < 0.0001 HR = 0.66	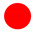
EXPAND [61], 2016	XP + Cetuximab	XP	EGFR	904	I	III	10.7 CTR arm9.4 EXP arm95%CI: 0.87–1.17 *p* = 0.95 HR = 1	5.6 CTR arm4.4 EXP arm95%CI: 0.92–1.29 *p* = 0.32 HR = 1.09	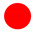

Nr: number; pts: patients; OS: overall survival; PFS: progression free survival; EXP: experimental; CTR: control; XELOX: capecitabine and oxaliplatin; EOC/EOX: epirubicin + oxaliplatin + capecitabine, XP: capecitabine and Cisplatin, FP: 5-fluorouracil and cisplatin, PTX: paclitaxel, CI: confidence interval, HR: hazard ratio. 
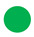
 positive study. 
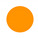
 partially positive study. 
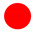
 negative study.

**Table 2 cancers-13-04094-t002:** Patient characteristics of the included studies (overall cohorts).

Author, Year, Acronym	Nr Total Pts (EXP/CTR)	Treatment Line (%)	Primary Endpoint	Setting	Molecular Target	EXP Arm	CTR Arm	Tumor Stage (%)	Tumor Site (%)	Notes
Li [69], 2016	267 (176/91)	III (65.1) IV(34.8)	OS, PFS	Adj	VEGFR-2	Apatinib	Placebo	III IV	Stomach (41.9) GEJ (13.5)	
Shah [74], 2017, METGastric	562 (279/283)	I (81.7) II (18.3)	OS	Adj	MET	Onartuzumab + FOLFOX6	Placebo + FOLFOX6	IV	Stomach (76.9) GEJ (23.1)	Enrollment was stopped early due to sponsor decision, which was agreed with the IDMC, due to a lack of efficacy in a phase II trial also assessing mFOLFOX6 plus onartuzumab
Thuss-Patience [32], 2017, GATSBY	345 (228/117)	II	OS	Adj	HER-2	Trastuzumab + Emtasine	Taxane	III (4) IV (95.9)	Stomach (68.1) GEJ (31.9)	
Catenacci [70], 2017RILOMET-1	609 (304/305)	I	OS	Adj	HGF	Rilotumumab + ECX	Placebo + ECX	IV (93.1) III (6.9)	Stomach (69.3) GEJ (20.4) Distal esophagus (10.3)	Study treatment was stopped early after a higher number of deaths in the rilotumab group.
Cunningham [71], 2017, UK Medical Research Council ST03	1063 (533/530)	I	OS	Periop	VEGF	Bevacizumab + ECX	ECX	Early (0.6) Advanced (91.7) Metastatic (0.19)	Stomach (55.7) GEJ (30.8) Distal esophagus (13.5)	EGJ type III was classified as gastric cancer
Fuchs [72], 2019, RAINFALL	645 (326/319)	I	PFS	Adj	VEGFR-2	Ramucirumab + Fluoropyrimidine + Cisplatin	Placebo + Fluoropyrimidine + cisplatin	IV (100)	Stomach (74.6) EGJ (25.3)	
Lorenzen [73], 2020, RADPAC	300(150/150)	II (57.7) III (31.7) IV (10.7)	OS	Adj	mTOR	Paclitaxel + Everolimus	Placebo + Paclitaxel	III IV	Stomach (41) GEJ (58.7)	
Shah [68], 2020 GAMMA-1	432 (218/214)	I	OS	Adj	MMP9	Andecaliximab + mFOLFOX6	Placebo + mFOLFOX6	IIIIV	Stomach (66) GEJ (34)	
Summary of Findings	4223 (2214/2009)	I (76) II (14.7) III (6.4) IV (3)	OS (87.5%) PFS (25%)	Adj 9 (87.5%) Periop 1 (12.5)					Stomach (63.5) EGJ (28.7) Esophagus (4.9)	2 trials were stopped early

Nr: number; pts: patients; Adj: adjuvant treatment; Periop: perioperative treatment; OS: overall survival; PFS: progression free survival; EXP: experimental; CTR: control; GEJ: gastroesophageal junction; FOLFOX6: fluorouracil leucovorin oxaliplatin; ECX: epirubicin, cisplatin, and capecitabine; mFOLFOX6: modified oxaliplatin, leucovorin, and fluorouracil; MMP9: matrix metalloproteinase 9.

**Table 3 cancers-13-04094-t003:** Characteristics of studies included in meta-analysis, along with information on primary outcomes. The positive (green dot) or negative (red dot) outcomes of each study are reported, consistent with its primary endpoint.

	OS	PFS	Results
Author, Year, Acronym	EXP	CTR	Nr	HR	Low	High	*p* Value	HR	Low	High	*p* Value	
Li [69], 2016	Apatinib	Placebo	267(176/91)	0.709	0.537	0.937	0.015	0.444	0.331	0.595	<0.001	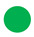
Shah [74], 2017, METGastric	Onartuzumab + FOLFOX6	Placebo + FOLFOX6	562(279/283)	0.82	0.59	1.15	0.24	0.90	0.71	1.16	0.43	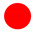
Thuss-Patience [32], 2017, GATSBY	Trastuzumab + Emtasine	Taxane	345(228/117)	1.15	0.87	1.51	0.86	1.13	0.89	1.43	0.31	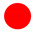
Catenacci [70], 2017, RILOMET-1	Rilotumab + ECX	Placebo + ECX	609(304/305)	1.34	1.10	1.63	0.003	1.26	1.04	1.51	0.016	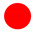
Cunningham [71], 2017, UK Medical Research Council ST03	Bevacizumab + ECX	ECX	1063(533/530)	1.08	0.91	1.29	0.36	1.05	0.89	1.23	0.56	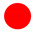
Fuchs [72], 2019, RAINFALL	Ramucirumab + Fluoropyrimidine + Cisplatin	Placebo + Fluoropyrimidine + Cisplatin	645(326/319)	0.962	0.801	1.156	0.68	0.753	0.607	0.935	0.011	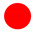
Lorenzen [73], 2020, RADPAC	Paclitaxel + Everolimus	Placebo + Paclitaxel	300(150/150)	0.93	0.73	1.18	0.544	0.88	0.70	1.11	0.273	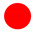
Shah [68], 2020 GAMMA-1	Andecaliximab + mFOLFOX6	Placebo + mFOLFOX6	432(218/214)	0.93	0.74	1.18	0.56	0.84	0.67	1.04	0.10	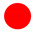

Nr: number; HR: hazard ratio; OS: overall survival; PFS: progression free survival; EXP: experimental; CTR: control; XELOX: capecitabine and oxaliplatin; FOLFOX6: fluorouracil leucovorin oxaliplatin; ECX: epirubicin, cisplatin, and capecitabine; mFOLFOX6: modified oxaliplatin, leucovorin, and fluorouracil; MMP9: matrix metalloproteinase 9. 
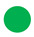
 positive study. 
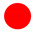
 negative study.

**Table 4 cancers-13-04094-t004:** Overall response rate and quality of life. This table summarizes the overall response rate based on RECIST criteria reported in the experimental and in the control arm for each study. The quality of life was reported according to EORTC QLQ-C30 questionnaire, which measure cancer patients’ physical, psychological, and social functions. This questionnaire is composed of multi-item scales and single items.

Author, Year, Acronym	Overall Response Rate			Quality of Life EORTC QLQ-C30
	EXP arm (%)	CTR arm (%)	*p* value	
Li [69], 2016	2.84	0.0	0.1695	No differences (*p* > 0.05)
Shah [74], 2017, METGastric	40.6	46.1	0.25	nd
Thuss-Patience [32], 2017, GATSBY	20.6	19.6	0.8406	nd
Catenacci [70], 2017, RILOMET-1	29.8	44.6	0.0005	nd
Cunningham [71], 2017, UK Medical Research Council ST03	41	42	0.70	nd
Fuchs [72], 2019, RAINFALL	41.1	36.4	0.17	HR 1.029 (0.786, 1.347)
Lorenzen [73], 2020, RADPAC	8	7.3	nd	nd
Shah [68], 2020, GAMMA-1	50.5	41.1	0.049	nd

EXP: Experimental; CTR: Control.

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
