# Peer review of "Molecularly Targeted Therapies for Gastric Cancer. State of the Art"

_cancers, 2021, doi:10.3390/cancers13164094_

Round 1

Reviewer 1 Report

The authors examined the efficacy and safety of novel drugs recently investigated. And the authors searched phase III randomized controlled trials. Then the authors concluded that the survival benefits of molecular therapies available to date for advanced and metastatic gastric cancer 36 are rather unclear.

This study is most recent meta-analysys of molecular therapy for gastric cancer. Unfortunately, only one study was positive in this review.

Selection methods and comparison strategies of these literatures were reasonable and clear. 

Author Response

1) response to reviewer 1 .

We thank  the first reviewer for his /her positive comments.

As concerns English language,  we  attach an American Journal Experts (AJE) certificate stating that the manuscript was edited for proper English language, grammar, punctuation, spelling, and overall style by one or more of the highly qualified native English speaking editors at AJE

Reviewer 2 Report

The authors retrace the relevant clinical trials with molecular therapies such as anti-epidermal growth factor receptor signaling including anti-HER2 and anti-EGFR1, anti-VEGF signaling, anti-mammalian target of rapamycin (mTOR), tyrosine kinase inhibitors (TKIs) and anti-MET performed in gastric cancer patients. While they are discussing the reasons for their failure due to the incomplete conduct of the same, or to the higher molecular heterogeneity of GC less attention is paid to perspective for a real improvement of  disease treatment.  The authors should include a paragraph on studies performed on patient-derived material (e.g. 3D cultures, PDX, PDOX), as tools to derive molecular information to better target GC for innovative MTs and stratify patients for clinical trials.

The review article is very well written, however the authors should address the following comments:

  • A list of abbreviations should be introduced (and control abbreviations to be properly introduced in text)
  • Fig 1 is not well executed: it is confusing and not very informative. I think that it should be better organized or removed
  • Almost all Figure legends are poorly described. The figure legend and tables should improved with information that makes them easier to interpret.
  • Standardize abbreviations
  • Fig3 lacks the legend for ! and + judgements, and domains are unclear. You can follow the example attached
  • Cite the method for bias analysis in the legend. Add ref number besides the studies (they lack also in other figures, i.e. fig4, 5, 6)
  • Lane 318-310: ref 67 and 73 must be inverted.
  • Too much empty space in fig 4, 5, S1, S2
  • Line 223/224 oesophagogastr* is repeated twice.
  • Line 390 The abbreviation QoL is already cited in lane 216.
  • Better explain (maybe in the legend to Table 4) what is EORTCQLQ-C30.

Author Response

1) Response to comment 1.  A paragraph on studies performed on patient-derived material  as tools to derive molecular information to better target GC for innovative MTs and stratify patients for clinical trials has been added at page 8.

2)Response to  further comments: 

  • A list of abbreviations should be introduced (and control abbreviations to be properly introduced in text)A list of abbreviations has been introduced at pages 2 and 3.  
  • Fig 1 is not well executed: it is confusing and not very informative. I think that it should be better organized or removedFigure 1 has been redone
  • Almost all Figure legends are poorly described. The figure legend and tables should improved with information that makes them easier to interpret.All figure and tables legends have been improved. 
  • Standardize abbreviations. All abbreviations have been standardized.
  • Fig3 lacks the legend for ! and + judgements, and domains are unclear. You can follow the example attached.Figure 3 has been improved following your suggestions.
  • Cite the method for bias analysis in the legend. The method for bias analysis has been cited in the legend.

         Add ref number besides the studies (they lack also in other figures, i.e.   fig. 4, 5, 6). The reference number has been added besides the studies in all included figures.

  • Lane 318-310: ref 67 and 73 must be inverted. References numbers have been completely modified.
  • Too much empty space in fig 4, 5, S1, S2. Figures 4, 5, S1, S2 have been modified.
  • Line 223/224 oesophagogastr* is repeated twice. The repeated word has been deleted.
  • Line 390 The abbreviation QoL is already cited in lane 216.The repeated abbreviation has been deleted.
  • Better explain (maybe in the legend to Table 4) what is EORTCQLQ-C30. EORTCQLQ-C30 was explained in Table 4 legend.